# Optimized Recovery of Cryostored Dormant Buds of Mulberry Germplasm

**DOI:** 10.3390/plants12020225

**Published:** 2023-01-04

**Authors:** Ravish Choudhary, Surendra Kumar Malik, Rekha Chaudhury, Atmakuri Ananda Rao

**Affiliations:** 1Seed Science and Technology, ICAR-Indian Agricultural Research Institute, Pusa Campus, New Delhi 110012, India; 2Tissue Culture and Cryopreservation Unit, ICAR-National Bureau of Plant Genetic Resources, Pusa Campus, New Delhi 110012, India; 3Central Sericultural Germplasm Resources Center, Hosur 635109, India

**Keywords:** *Morus* spp., cryopreservation, two-step freezing, dormant buds, in vitro regeneration, recovery

## Abstract

A two-step freezing cryoprotocol preceded by desiccation to 15 to 25% moisture content was developed and successfully applied to winter dormant buds of mulberry (different *Morus* spp.) of a core set comprising 238 accessions studies in our laboratory. The survival and recovery percentage of diverse accessions cryobanked for various periods were tested under in vitro conditions, and several factors were analyzed to determine their role in optimizing the recovery of low-viability accessions. The effect of rates of freezing and thawing (both fast and slow), were tested and recovery compared. Recovery conditions such as dark incubation and rehydration in sterile moist moss grass for different durations after cryopreservation led to a higher survival percentage compared to controls. Two different recovery culture media were compared for their efficiency in survival. On average, the survival under in vitro culture conditions using optimized conditions was high: above 60% in majority of the accessions. Dormant buds showed viability in the range of 25 to 100% with an average of 50.4%. The recovery percentage of winter dormant buds after cryopreservation via slow freezing and slow thawing with rehydration by moist moss grass for 2 h was recorded in the range from 63.3 to 90.9% with an average of 81.05%. Without rehydration, it ranged from 50 to 75% with an average of 60.4%. Regeneration of cryopreserved mulberry germplasm after 6 years of storage indicated no survival loss over different years of storage, and 33–40% of the accessions showed viability above 40%, up to a maximum of 100%. Maximum shoot formation (100%) was obtained from *Morus alba*. The majority of the accessions were rooted in vitro within 20–25 days of subculture in the auxin rich rooting media, except in wild species *M. latifolia* and *M. laevigata*, which took longer (45 to 60 days) for root development. All the rooted plantlets were then transferred to the field and successfully established in a glasshouse.

## 1. Introduction

Mulberries are one of the healthiest foods among horticulture crops, and there has recently been an increase in interest in less well-known horticultural plant species such as these. Mulberry fruits are scrumptious, healthy, and offer a host of remarkable health advantages. Less-known fruits are accepted as insurance in the future horticultural cultivation in the climate-change scenario due to their resistance to abiotic and biotic conditions [1,2,3]. Mulberry (*Morus* L., Family *Moraceae*) that originated at the foothills of the Himalayas is an economically important woody tree species which is extensively used for rearing silkworms. It is an out-breeding, heterozygous, and perennial tree species also used in agro-forestry and horticulture [4,5,6]. Sanjappa [7] recognized 68 species within the genus *Morus* and in India, 4 main species of *Morus*, viz., *M. indica*, *M. alba*, *M. laevigata*, and *M. serrata* have been reported [8]. The Central Sericultural Germplasm Resources Center (CSGRC), Hosur, Tamil Nadu, (India) is the National Active Germplasm site for mulberry germplasm in India, which conserves 13 *Morus* species from 26 different countries, totaling 1120 accessions in their field gene-bank [9]. The cryopreservation of dormant buds of mulberry germplasm as an economical, safe, and effective method of long-term conservation has been reported as a safety back up of field gene-banks. The cryopreservation has proved its worth for ensuring for long-term conservation and utilization of plant genetic resources [10,11,12,13,14]. The majority of wild *Morus* species showed sensitivity in nature and demonstrated species-specific variation under in vivo and in vitro conditions [14]. Some wild mulberry germplasms such as *M. tiliaefolia* and *M. serrata* showed significantly less recovery after cryopreservation [14,15]. However, success can be judged only when high recovery rates are achieved after cryostorage. At the National Bureau of Plant Genetic Resources (NBPGR), New Delhi, 238 accessions of the core set of mulberry germplasm are maintained at National Cryogenebank in the form of dormant buds from −170 °C to −180 °C temperature [15]. These buds need to be recovered at the times of need for raising complete plantlets. In the present study, the cryobanked winter dormant buds of mulberries were recovered using different variables to optimize the recovery conditions via an in vitro culture method for subsequent field transfer.

## 2. Results

The effect of various variables, as tested by each individually or in combination, was recorded in terms of recovery percentages.

### 2.1. Effect of Harvest Dates

Buds of more than 50 accessions harvested during the months of December, January, and February for three consecutive years were checked for initial moisture contents. Moisture content was found to range from 24.2 to 61%. Accessions were categorized into each of the moisture content (MC) ranges and expressed as percentage values (Figure 1). It was observed that a large proportion (91%) of December harvested buds had MC range values of 36–55% and 9% showed high moisture content (above 55%). January harvested buds also showed similar results, but 5% of the samples showed a slightly lower value of moisture content (26–35%). More than 95% of February-harvested buds showed moisture content in the range from 26 to 55%. Temperatures during December, January, and February months ranged from a maximum of 25–27 °C to a minimum of 14–15 °C and nil or negligible rainfall. Based on these observations in subsequent experiments, buds were harvested only during mid-February for cryopreservation.

### 2.2. Relation between Desiccation Level and Survival of Buds after Cryopreservation

Moisture content of freshly harvested dormant buds of different Morus species ranged from 28.1 to 59.6% with an average of 47.5%, and initial viability was found in the range from 46.7 to 100% with an average of 75.1%. Moisture content after 4–5 h desiccation in charged silica gel ranged from 9.5 to 17.8%, with an average of 15.2% observed in the present study. Winter dormant buds after removal of 5–8 outer scales were tested for viability before and after cryostorage. The viability of dormant buds before cryopreservation after desiccation ranged from 25 to 100% with an average of 51.5%. After cryostorage, no significant change in the viability was noticed. Dormant buds showed viability in the range of 25 to 100% with an average of 50.4%. Maximum viability (100%) was found after cryopreservation in the accessions of *Morus indica* × *M. alba* (IC 313836) at 17.3% of moisture content after 5 h desiccation followed by *M. indica* (IC 313703) (95%) at 15.31% moisture content and *M. alba* (EC 493822) (80%) at 16.11% moisture content. Minimum viability (25%) was recorded in *M. rubra* (EC 493988) at 15.5% moisture content, followed by *M. latifolia* (EC 493831) with viability (28.6%) at 9.5% moisture content and *M. cathayana* (EC 493775) with viability (30%) at 13.8% of moisture content (Table 1). The seedlings obtained from cryostored dormant buds were visually normal and healthy. A total set of core collection of 13 diverse species of mulberry germplasm from different parts of India have been successfully cryostored as a base collection in the cryogenebank (Figure 2).

### 2.3. Effect of Slow, Fast, and Two-Step Cooling Rate

The recovery percentage of winter dormant buds after cryopreservation through slow freezing and slow thawing with rehydration by moist moss grass for 2 h was recorded in the range of 63.3 to 90.9% with an average of 81.1%. Without rehydration, it ranged from 50 to 75% with an average of 60.4%. The recovery was found through slow freezing and fast thawing with rehydration in the range of 7.7 to 52.4% with an average of 35.8%. Without rehydration, this ranged from 0 to 33.3% with an average of 18.7%. However, the recovery of dormant buds after cryostorage through fast freezing and slow thawing with rehydration was obtained in the range from 11.1 to 36.4, with an average of 23.8%. Without rehydration, this ranged from 3.3 to 16.7% with an average of 10.7%, while recovery showed fast freezing by fast thawing with rehydration in the range of 6.7 to 25.4% with an average of 16.3%. Without rehydration, this ranged from 0 to 16.7% with an average of 7.5%. The highest recovery (90.9%) was found in the accession of *M. indica* (IC 313887) after cryopreservation by slow freezing and slow thawing with rehydration in moist moss grass for 2 h at room temperature. Without rehydration, recovery was found 75% in *M. alba* (IC 313736). In the case of slow freezing followed by fast thawing, the higher recovery was found 52.4% with rehydration and 33.3% recovery was observed without rehydration in *M. indica* (IC 313977). However, in case of fast freezing followed by slow thawing, the maximum recovery was recorded 36.4% with rehydration in *M. indica × M. alba* (IC 493875), while without rehydration, it was found 16.7% in *M. indica*. Meanwhile, 25.3% recovery was found after fast freezing followed by fast thawing and rehydration, and without rehydration, the recovery was 16.7% in *M. indica × M. alba* (IC 493875) (Table 2).

### 2.4. Effect of Thawing and Rehydration

The viability percentage of dormant buds was found to be at its maximum between moisture contents of 15–25%. Above 25% and below 15% of moisture content, the viability of buds gradually decreased in cases with slow freezing and fast freezing. In the case of fast freezing using fast thawing, the viability of buds above 50% of moisture content was lost. However, during slow freezing, a maximum 80% viability was observed, while in case of fast freezing, 70% was found at 19% moisture content (Figure 3).

### 2.5. Effect of Recovery Media

The percentage of shoot formation ranged from 26.7 to 100% with an average of 71.7% in TDZ (0.1 mgL^−1^) while in case of BAP (1 mgL^−1^), it was between 33.3 and 100% (Table 3). The viability was a maximum of 100% with the help of BAP in M. indica (IC 314255) and the minimum was 33.3% in M. alba (IC 314257) while for TDZ, maximum viability (100%) was obtained in M. indica (IC 314255) and the minimum was 26.7% M. alba (IC 314257). With the exception of two accessions, IC 405829 (M. indica) and IC 405800 (M. bombycis), BAP (1 mgL^−1^) generally outperformed TDZ (0.1 mgL^−1^) in terms of viability percentage, with IC 405829 (M. indica) and IC 405800 (M. bombycis) showing much higher viability than BAP (1 mgL^−1^). After retrieval, the buds started sprouting in vitro after 3–4 week of culturing (Figure 4A). Once the bud sprouting started, the shoot formation, rooting, and hardening (Figure 4E–H) was normal. After elongation, complete plantlets were transferred for hardening in the field condition. The initial growth of frozen meristems was extremely slow compared to unfrozen buds. However, leaves began to expand within 2 weeks and shoot development took place during 3–4 weeks in culture. In the current study, the shoots derived from cryopreserved winter buds produced roots in MS medium supplemented with 0.1 mgL^−1^ NAA and, after hardening in pots filled with vermiculite and peat moss (2:1) in a humified atmosphere, they were transferred to soil.

### 2.6. Effect of Dark Incubation

Buds that did not show any recovery following direct culturing under light condition showed viability of 12–100% with an average of 52% in all the different *Morus* species when culture conditions were modified by dark incubation. The highest recovery (100%) was recorded after modified conditions in *M. latifoila* (EC 493779), while minimum was 12% in *M. alba* (EC 493799) (Table 4).

### 2.7. Viability after Cryostorage

The viability percentage was initially found to be 57.7% (the average of eight accessions of different *Morus* species) after 3 months of cryostorage. However, no significant loss was observed in terms of viability percentage after 6 years of cryostorage, which was found to be 54.2% (Table 5).

## 3. Discussion

Dormant buds are subjected to cold acclimation. Plants exposed to naturally cold, non-freezing temperatures have high tolerance to subsequent artificial freezing, enabling them to be cryoconserved, ensuring long-term viability. Several physiological and biochemical mechanisms are known to confer this natural tolerance. Difference in hardiness to low temperatures in nature is observed between shoot tissues, with bud and cortex being the most hardy tissue and xylem being the least hardy [16]. Tolerance of such acclimated dormant buds to a much lower temperature of liquid nitrogen, however, depends on extracellular freezing, which is caused by slow stepwise freezing. This was studied in mulberry [14,17,18,19], in almond [20], in apple [21], in pear, in elm [13], etc. The techniques, however, so devised ultimately depend on successful higher recovery.

Most species are reliably preserved after being desiccated to 20-30% [22] and then cooled slowly between −30 and −40 °C before exposure to the vapor phase of liquid nitrogen [16,23,24]. In the present study, *Morus indica*, *M. alba*, *M. bombycis* and *M. sinensis* showed high growth values when pre-desiccated to moisture contents between 15 and 25%. After two-step freezing, high viability was observed. It was also observed that most of the accessions, which survived LN exposure, had inherent dormant or semi-dormant nature, whereas accessions that failed to recover after LN exposure were non-dormant or semi-dormant in nature (Table 1). It seems there is a correlation between the dormancy of the plant and freezing tolerance leading to recovery after LN exposure. The values of moisture content and in vitro survival fresh, after desiccation and cryoexposure were shown in Table 1.

Tyler and Stushnoff [21] mentioned that the desiccation rates are significantly correlated to the size and compactness of the buds. Niino et al. [25] found that in winter mulberry buds, with small vascular tissues, partial dehydration prior to pre-freezing improved shoot formation in meristem culture of cryostored buds after thawing. In *Malus* species, high survival can be obtained with nodal segments, whereas the moisture content is reduced from 45% (moisture content when collected) to 25 to 30% prior to cooling [26]. Pear buds had the best survival at 41% [27]. In our study, dormant buds of *Morus* species showed higher survival when the moisture content was reduced by 15 to 25%. As well as increasing the desiccation, the buds decreased the moisture content and viability before and after cryopreservation.

The last pre-freezing temperature (just before plunge in liquid nitrogen) required for survival after cryostorage depends on the thawing method. Suzuki et al. [27] found that higher rates of shoot formation were obtained at −30 °C pre-freezing temperature, followed by slow thawing in the pear germplasm, and similarly in apple [21,28]. However, in mulberry, higher survival was obtained at −20 °C, followed by rapid thawing in a ±37 °C water bath (3–5 min) [29]. The simplest and quickest pretreatment for the successful cryopreservation of mulberry buds was a 24 h exposure of the samples to each lowering temperature (−5 °C/day) before being plunged in liquid nitrogen and subsequent rapid thawing ±38 °C [29]. During freezing, water migrates from the super cooled shoot primordial tissue to the bud scales and the dehydrated shoot tips attain a high degree of freezing tolerance, preventing intracellular freezing and lethal ice nucleation [10]. Pre-freezing rate in combination with the last freezing temperature as well as thawing methods was equally factors in improving the shoot formation rates of cryopreserved buds. Pre-freezing with a daily decrease of −5 °C in temperature gave good results compared to decreases of −10 °C [30]. It is important to dehydrate the tissues to prevent lethal intracellular cooling prior to rapid freezing to produce higher survival after cryopreservation [18]. Yakuwa and Oka [29] also reported that meristems excised from segments without pre-freezing and transferred directly from 0 to −196 °C never survived.

In our study, attempts were made to compare the role of thawing in recovery rates of buds cryopreserved by two rates of cooling, namely, slow and fast. Data obtained for accessions of different *Morus* species are shown in Table 2. Buds conserved by slow freezing followed by slow thawing with rehydration showed a high recovery percentage (avg. 81.1%), while 60.4% recovery was recorded without rehydration. Using fast thawing for buds conserved by slow freezing led to a lower percent recovery of 35.8% and 18.7%, respectively. In contrast, using fast freezing, slow thawing led to recovery of 23.8% and 10.7% for rehydration and non-rehydrated buds, respectively. However, using fast freezing and fast thawing with rehydration recovery, the percentage was very low (avg. 16.3%). Such buds without rehydration had much lower percentages at 7.5% (Table 2).

Touchell and Walters [31] found that embryos of *Zizania palustris* were rehydrated for 60 min. on moist filter paper to reduce imbibitional damage after cryopreservation. Thawed buds were rehydrated for 2–4 h in sterile moist moss grass to reduce imbibitional injury. In our study, high survival was obtained with rehydration in moist moss grass in compared to a lack of rehydration (Table 2). Apparently, slow thawing for stepwise freezing and fast thawing for fast frozen samples are the optimal methods for optimizing recovery. Similarly, recovery conditions including dark incubation and rehydration in sterile moist moss grass for different durations led to higher survival in dormant buds of almond [32] and Himalyan mulberry [33].

Conventional methods of propagation of mulberry through cuttings and grafting have certain limitations. Bapet et al. [34] reported that only 30–40% of stem cuttings survive the time period between pruning, transportation, and final transplantation. Ohyama and Oka [35] found that rooting from stem cutting depends on the environmental factors and physiological state of the cuttings, while grafting depends on the internal factors such as compatibility, nutrient and moisture content of the scion, activity of the cambium, as well as external factors such as atmospheric temperature and soil mixture [36]. In vitro propagation has been utilized for large-scale propagation of several tree species [37,38,39,40,41]. The regeneration of complete plantlets in vitro from apical/axillary shoot buds or nodal explants [35,42,43,44] and from cryostored buds [15,29,45] have been reported in several mulberry species. According to Ivanicka [43], MS medium [46] is the most efficient for mulberry species. In the present study, shoot formation was observed faster in case of 0.1 mgL^−1^ TDZ but survival percent was found to be at its maximum in case of 1 mgL^−1^ BAP. Similarly, thidiazuron (TDZ) at 0.1 mgL^–1^ or benzylaminopurine (BAP) at 1.0 mg L^–1^, added separately, gave the best rates of shoot initiation and shoot induction, and required less time for bud sprouting in *M. alba*, *M. indica*, and *M. laevigata* [47].

Low survival and shoot formation in *Morus laevigata*, *M. serrata*, *M. tilliaefolia*, *M. rubra*, *M. rotundiloba*, and *M. cathayana* showed a need for modification of culturing conditions [48]. In the present study, *M. cathayana*, *M. rubra*, and *M. serrata* struggled to survive. Khurana et al. [49] found that MS medium supplemented with BAP (1 mgL^−1^) and low concentration of GA_3_ (0.1 mgL^−1^) and NAA (0.2 to 0.5 mgL^−1^) increased shoot elongation after cryopreservation.

Exposure of tissues to light during early recovery after liquid nitrogen exposure increased damage in shoot apices of *Solanum tuberosum* [50]. Because freezing and desiccation stresses are associated with cryostorage, tissues exposed to liquid nitrogen may be predisposed to other stresses, such as photo oxidative stress, during thawing and recovery [51,52]. Elstner et al. [53] found when plant tissues are exposed to stress, susceptibility to photo oxidative damage is increased. Based on the hypothesis that cryoexposure of tissues predisposes them to free radical attack, it is proposed that post-thaw conditions can influence oxidative damage and further reduce survival. In our study, higher survival was achieved with dark incubation. Buds that did not show any recovery on direct culturing under light conditions showed viability of 12–100% (Table 4). Overall, the current study showed that for *Morus* species, no significant decrement of survival was observed after 6 years of cryostorage at −196 °C at National Cryogenebank, NBPGR, New Delhi (India). Niino et al. [10] also demonstrated that no significant decrease was observed after 5 years of storage at −135 °C. Similarly, Rao et al. [14,15] also reported that the viability percentage of different *Morus* species was retained after 3 years of cryopreservation and was not altered.

## 4. Materials and Methods

Twigs about 50 cm long of mulberry of specified germplasm were collected from the field genebank of CSGRC, Hosur, Tamil Nadu, during the months of December, January, and February, wrapped in cotton bags, and air lifted to cryolab at NBPGR. Winter dormant buds were harvested from 1-year-old lateral shoots of mature trees. Winter buds were cryopreserved in liquid nitrogen using partial desiccation followed by a two-step freezing protocol, as described earlier [15,19,20,32,33].

Seven variables were tested for optimization of recovery growth.

Twigs harvested at regular intervals, during 2nd week of December to 2nd week of February, from a field genebank were examined for their moisture contents. Buds, after removing 5–7 outer scales, were pre-desiccated for periods ranging from 4–7 h over silica gel at room temperature. The moisture content (MC %) of fresh and desiccated dormant buds was determined via drying at 103 ± 2 °C in an oven for 17 h and expressed on a fresh weight basis. Dormant buds pre-desiccated to varying moisture contents were subjected to slow or fast freezing after being enclosed in 1.8 mL cryovials. The slow stepwise freezing was achieved by sequentially lowering the temperature by −5 °C/day using deep freezers up to a terminal temperature of −30 °C before plunging in liquid nitrogen at −196 °C. Fast freezing was achieved by direct plunging in liquid nitrogen.

Thawing was achieved using 2 methods: slowly, by keeping LN-retrieved cryovials in air at ambient temperature for about 40 min; or fast, by plunging in a water bath maintained at 38 °C for 3–5 min. Differently treated buds before culturing in vitro were rehydrated by sterile moist moss grass for 2 h at room temperature and the results compared with those cultured without rehydration. The retrieved buds were placed in vitro after further removal of 2 outer scales and washing with Tween20 for 15 min, followed by continuous water washing for 10 min. Buds were then surface sterilized with sodium hypochloride for 9 min and rinsed repeatedly three times with sterile autoclaved distilled water (5 min. each). The sterilized buds were then cultured on MS medium with 3% sucrose (*w*/*v*) and solidified with 0.8% agar. MS medium was initially supplemented with 1 mgL^−1^ 6-benzylaminopurine (BAP) and 0.1 mgL^−1^ thidiazuron (TDZ) for bud sprouting. After bud sprouting, they were subcultured in the MS medium supplemented with 1 mgL^−1^ BAP and 0.1 mgL^−1^ GA_3_ for elongation.

The in vitro cultured buds were incubated under complete darkness in a culture room at 25 ± 2 °C for 7 days. Tubes were then transferred under dim light by remaining on upper racks/shelves of the culture trolley for 3 days. Cultures were finally transferred to normal light conditions of 65 ± 10 µmol m^−2^ s^−1^ with a 16 h photoperiod in in vitro culture room.

### Statistical Analysis

A total of 25 explants were used in all the experiments with three replications, and the mean values were considered. Standard errors (SE) of the arithmetic means were calculated for each experiment. The data were analyzed by analysis of variance (ANOVA) in SPSS software for Windows (Release 15.0; SPSS Inc., Chicago, IL, USA). Significant differences between means were assessed using Duncan’s multiple range test (DMRT) at *p* ≤ 0.05.

## 5. Conclusions

The present study is the first report of physiological studies and cryostorage of mulberry dormant buds harvested from plants grown in subtropical conditions where temperatures do not fall below 12 °C in any season. The different factors were investigated to influence recovery conditions. In temperate species of apple, pear, etc., the buds have been recorded to be harvested only after plants have been exposed to subzero temperatures for a minimum of 72 h. In this study, in which dormant buds have been harvested from subtropical zone, there is no such consideration. Hence, moisture levels in buds for three different winter months, namely December, January, and February, when leaves were shed, were used as criteria. During acclimatization, buds are reported to lose their water content to different extents. In our study, the buds harvested in February showed the lowest moisture content values in comparison to those harvested earlier or later, and hence were found to be optimal. Although mulberry plants grown at CSGRC, Hosur, are not exposed to any cold in nature, there is a decline in moisture content in a certain growth period which can be equated to maximum dormancy or cold acclimation, although not fully.

## Figures and Tables

**Figure 1 plants-12-00225-f001:**
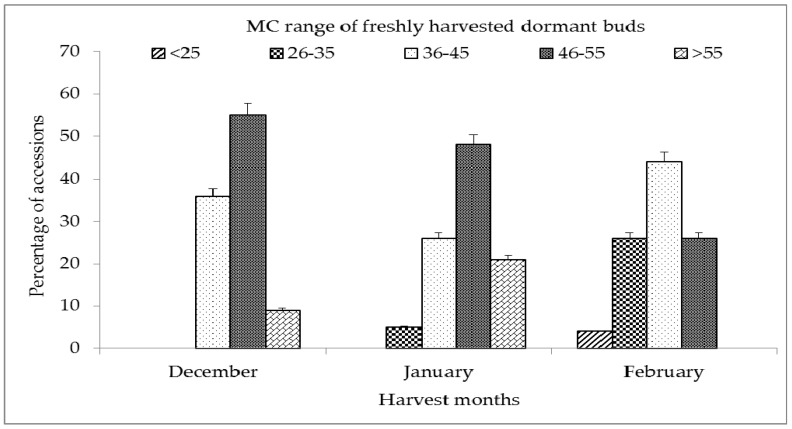
Moisture content (MC%) of dormant buds of different mulberry germplasm harvested in three winter months. MC% values are cumulative of 3 years data. Error bars showing mean values ±SD (n = 3).

**Figure 2 plants-12-00225-f002:**
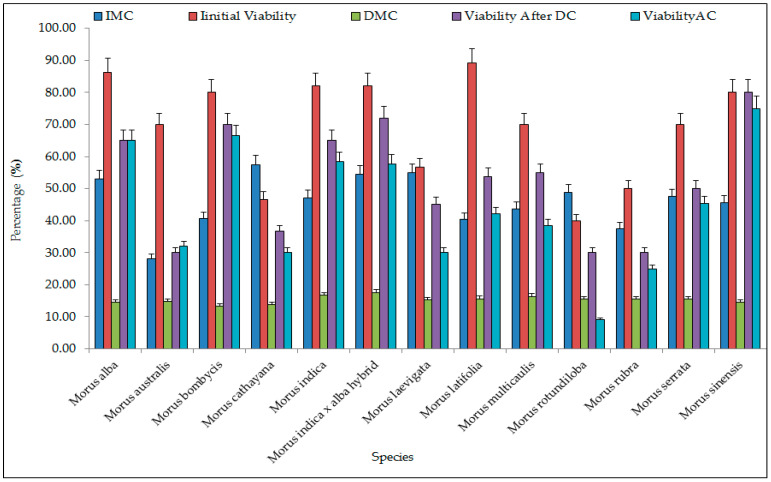
Viability of dormant buds of different *Morus* species after cryostorage. Error bars showing mean values ± SD (n = 3). (IMC = initial moisture content; DMC = moisture content after desiccation; DC = desiccation; AC = after cryopreservation).

**Figure 3 plants-12-00225-f003:**
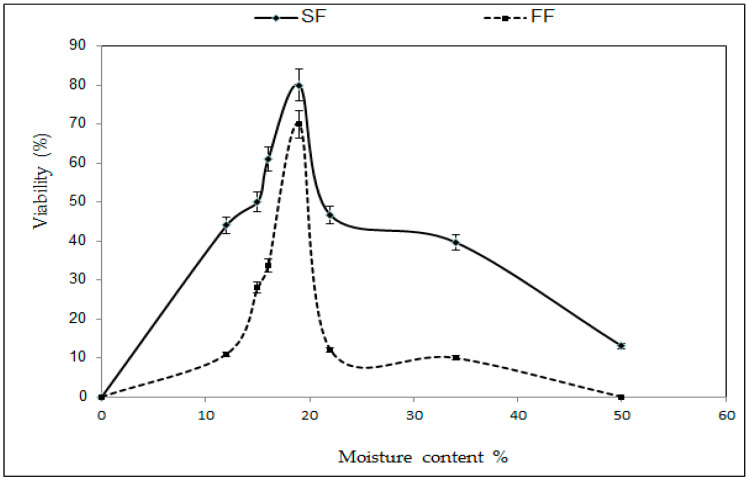
Relationship between moisture content% and viability% after slow freezing (SF) and fast freezing (FF) using fast thawing in *M. indica.* Error bars showing mean values ±SD (n = 3).

**Figure 4 plants-12-00225-f004:**
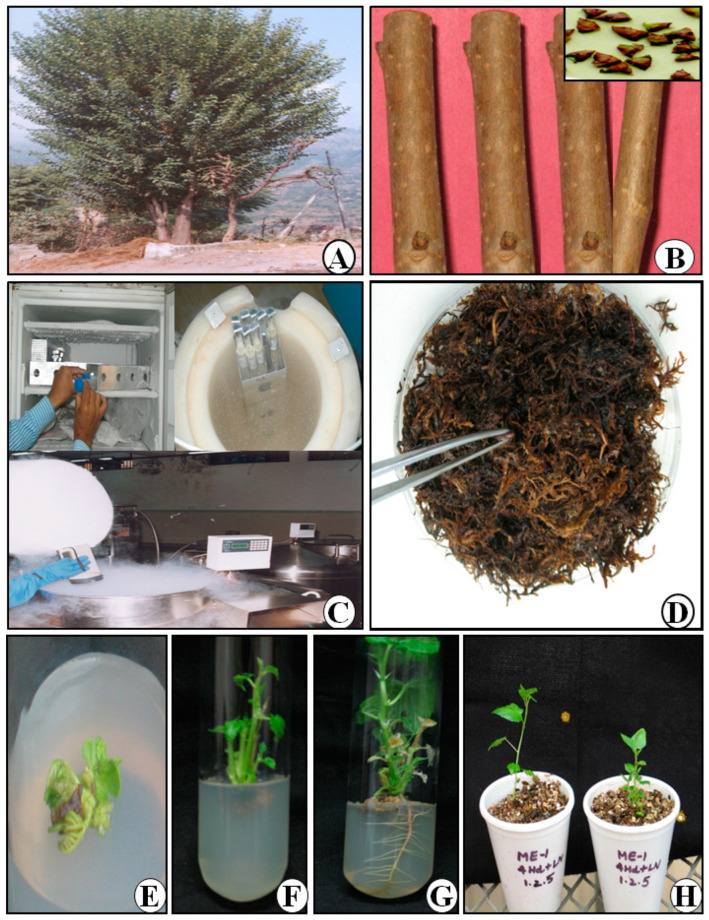
Recovery growth of mulberry from cryopreserved dormant buds after using different critical factors. (**A**) Mulberry tree in field genebank in winter period, (**B**) explants as a twig or dormant buds for cryopreservation, (**C**) freezing at different temperature and finally plunged in cryotank, (**D**) rehydration in sterile moist moss grass for different duration, (**E**) in vitro recovery of cryopreserved dormant buds of different Morus species, in vitro sprouting of dormant buds after cryostorage, (**F**) shoot multiplication and elongation, (**G**) rooting, (**H**) hardening of in vitro recovered plant. All photos were taken by the authors and all photos are original.

**Table 1 plants-12-00225-t001:** Viability and dormancy status of dormant buds of different Morus species.

S. No	Species	Accession No.	Initial Moisture Content (%)	Initial Viability (%)	Desiccated Moisture Content (%) (DMC)	Viability (%) at DMC	Viability (%) after Cryo-Exposure	Status
1.	*Morus alba*	EC 493822	51.2 (±0.55) ^b^	90.0 (±4.08) ^b^	16.1 (±0.42) ^c^	80.0 (±2.36) ^b^	80.0 (±2.36) ^b^	SD
2.	*Morus australis*	EC 493758	28.1 (±0.44) ^e^	70.0 (±4.08) ^c^	14.8 (±0.41) ^e^	30.0 (±4.71) ^g^	32.0 (±2.72) ^e^	SD
3.	*Morus bombycis*	EC 493785	40.7 (±0.47) ^cd^	80.0 (±2.36) ^bc^	13.3 (±0.38) ^f^	70.0 (±2.36) ^c^	66.5 (±3.60) ^c^	D
4.	*Morus cathayana*	EC 493775	57.5 (±0.47) ^ab^	46.7 (±2.72) ^f^	13.8 (±0.48) ^f^	36.7 (±2.72) ^g^	30.0 (±2.72) ^e^	SD
5.	*Morus indica*	IC 313697	52.4 (±0.47) ^b^	60.0 (±2.36) ^d^	15.3 (±0.04) ^d^	45.0 (±2.36) ^e^	40.0 (±2.36) ^d^	ND
6.	*Morus indica*	IC 313703	45.5 (±0.56) ^c^	95.0 (±0.00) ^ab^	15.3 (±0.51) ^d^	80.0 (±2.36) ^b^	95.0 (±2.36) ^ab^	ND
7.	*Morus indica*	IC 313711	51.8 (±0.24) ^b^	65.0 (±2.36) ^d^	15.6 (±0.28) ^d^	50.0 (±4.08) ^e^	40.0 (±4.71) ^d^	ND
8.	*Morus indica*	IC 313741	59.6 (±0.32) ^a^	60.0 (±2.36) ^d^	17.4 (±0.01) ^b^	40.0 (±4.71) ^f^	35.0 (±4.08) ^e^	ND
9.	*Morus indica*	IC 313843	51.5 (±0.65) ^b^	90.0 (±2.36) ^b^	17.8 (±0.18) ^b^	80.0 (±4.08) ^b^	80.0 (±4.71) ^b^	ND
10.	*Morus indica* × *alba*	IC 313992	56.9 (±0.61) ^ab^	100.0 (±0.00) ^a^	17.3 (±0.43) ^b^	100.0 (±0.00) ^a^	100.0 (±0.00) ^a^	ND
11.	*Morus indica* × *alba*	IC 313836	49.5 (±0.26) ^bc^	70.0 (±4.71) ^c^	16.6 (±0.14) ^c^	60.0 (±4.71) ^d^	40.0 (±4.08) ^d^	ND
12.	*Morus laevigata*	IC 313789	56.8 (±0.69) ^ab^	70.0 (±4.08) ^c^	15.3 (±0.32) ^d^	50.0 (±2.36) ^e^	40.0 (±4.08) ^d^	D
13.	*Morus latifolia*	EC 493817	40.2 (±0.11) ^cd^	100.0 (±0.00) ^a^	23.6 (±0.37) ^a^	80.0 (±4.71) ^b^	39.7 (±4.95) ^d^	SD
14.	*Morus latifolia*	EC 493823	51.6 (±0.47) ^b^	70.0 (±4.71) ^c^	14.9 (±0.09) ^e^	46.7 (±5.44) ^e^	40.0 (±4.71) ^d^	SD
15.	*Morus latifolia*	EC 493831	36.7 (±0.48) ^d^	100.0 (±0.00) ^a^	09.5 (±0.15) ^g^	25.0 (±0.91) ^h^	28.6 (±1.36) ^ef^	D
16.	*Morus multicaulis*	EC 493763	43.7 (±0.74) ^c^	70.0 (±4.08) ^c^	16.3 (±0.01) ^c^	55.0 (±2.36) ^de^	38.4 (±3.60) ^d^	SD
17.	*Morus rubra*	EC 493988	37.6 (±0.54) ^d^	50.0 (±4.71) ^e^	15.5 (±0.05) ^d^	30.0 (±2.36) ^g^	25.0 (±2.36) ^f^	D
18.	*Morus serrata*	IC 314167	45.4 (±0.47) ^c^	50.0 (±4.71) ^e^	15.4 (±0.01) ^d^	40.0 (±4.71) ^f^	32.0 (±2.72) ^e^	D
19.	*Morus sinensis*	IC 313974	45.5 (±0.52) ^c^	80.0 (±4.71) ^bc^	14.5 (±0.24) ^e^	70.0 (±2.36) ^c^	75.0 (±2.36) ^bc^	ND
	Average		47.5 (±0.48)	75.1 (±2.91)	15.2 (±0.24)	51.5 (±2.73)	50.4 (±3.15)	

ND—non-dormant; SD—semi-dormant; D—dormant. All the values are the average ± SD of three replicates (n = 3), and the means with the same letter (superscript) in the columns are not significantly different (*p* < 0.05)—(Duncan’s multiple range test).

**Table 2 plants-12-00225-t002:** Recovery percentage for dormant buds of different *Morus* species cryoconserved using slow and fast freezing after partial desiccation with and without rehydration.

Accession No	Recovery Percentage (%)							
	Slow Freezing (2-Step Freezing)				Fast Freezing (Direct Plunging in LN)			
	Slow Thawing with Rehydration	Slow Thawing without Rehydration	Fast Thawing with Rehydration	Fast Thawing without Rehydration	Slow Thawing with Rehydration	Slow Thawing without Rehydration	Fast Thawing with Rehydration	Fast Thawing without Rehydration
*M. indica* × *alba* [IC 493875]	63.3 (±2.72) ^c^	50.0 (±4.71) ^c^	41.7 (±4.91) ^b^	25.0 (±2.36) ^b^	36.4 (±2.72) ^a^	12.5 (±1.18) ^ab^	25.3 (±0.47) ^a^	16.7 (±0.33) ^a^
*M. alba* [IC 313736]	80.0 (±2.36) ^b^	75.0 (±2.36) ^a^	07.7 (±0.14) ^c^	00.0 (±0.00)	14.3 (±0.01) ^b^	10.2 (±0.42) ^b^	06.7 (±0.48) ^c^	00.0 (±0.00)
*M. indica* [IC 313977]	90.0 (±4.71) ^a^	66.7 (±2.72) ^b^	52.4 (±0.48) ^a^	33.3 (±2.72) ^a^	11.1 (±0.47) ^b^	03.3 (±0.27) ^c^	13.1 (±1.36) ^b^	06.7 (±0.27) ^b^
*M. indica* [IC 313887]	90.9 (±4.09) ^a^	50.0 (±4.71) ^c^	41.7 (±1.36) ^b^	16.7 (±2.72) ^c^	33.3 (±1.22) ^a^	16.7 (±2.72) ^a^	20.0 (±4.71) ^ab^	06.7 (±0.27) ^b^
Average	81.1 (±3.47)	60.4 (±4.83)	35.8 (±1.72)	18.7 (±1.95)	23.8 (±1.10)	10.7 (±1.15)	16.3 (±1.70)	7.5 (±0.22)

All the values are the average of three replicates (n = 3), and the mean with the same letter (superscript) in the columns are not significantly different (*p* < 0.05)—(Duncan’s multiple range test).

**Table 3 plants-12-00225-t003:** Effect of different plant growth regulators (PGRs) on viability percentage of dormant buds of different *Morus* species.

PGR	*M. alba* IC 314257	*M. bombycis* IC 405829	*M. indica* IC 314248	*M. indica* IC 405817	*M. indica* IC 314247	*M. indica* IC 314094	*M. indica* IC 405800	*M. indica* IC 314255	Avg. Viability (%)
BAP (1 mgL^−1^)	33.3 (±2.72) ^a^	73.3 (±2.72) ^b^	73.3 (±5.44) ^a^	83.3 (±6.80) ^a^	93.3 (±2.72) ^a^	70.0 (±4.71) ^a^	86.7 (±5.44) ^b^	100.0 (±0.00)	76.6
TDZ (0.1 mgL^−1^)	26.7 (±2.27) ^b^	96.7 (±2.72) ^a^	66.7 (±2.72) ^b^	73.3 (±2.72) ^b^	86.7 (±5.44) ^b^	63.3 (±2.72) ^b^	90.0 (±4.71) ^a^	100.0 (±0.00)	75.4

All the values are the average of three replicates (n = 3), and the mean with the same letter (superscript) in the columns are not significantly different (*p* < 0.05)—(Duncan’s multiple range test).

**Table 4 plants-12-00225-t004:** Improved viability of cryostored *Morus* species after rehydration and culturing in the dark.

S. No	Species	Accession No.	Viability (%)	
			Direct Culturing *	After Modifying Culture Conditions **
1	*Morus alba*	EC 493799	0.00	12.0 (±1.63) ^d^
2	*Morus bombycis*	EC 493821	0.00	20.0 (±4.71) ^cd^
3	*Morus cathayana*	EC 493775	0.00	30.0 (±4.71) ^c^
4	*Morus latifolia*	EC 493779	0.00	100.0 (±0.00) ^a^
5	*Morus latifolia*	EC 493819	0.00	75.0 (±2.36) ^b^
6	*Morus sinensis*	IC 313974	0.00	75.0 (±2.36) ^b^

* Buds descaled and inoculated in vitro; ** buds descaled then rehydrated in 100% RH for 2 h and then cultured for 7 days in complete dark and 4 days in dim light. All the values are the average, ± SD of three replicates (n = 3), and the means with the same letter (superscript) in the columns are not significantly different (*p* < 0.05)—(Duncan’s multiple range test).

**Table 5 plants-12-00225-t005:** Viability of mulberry germplasm after different duration of cryostorage.

S. No	Species	Accession No.	Viability (%) after Different Duration of Cryostorage	
			(3 Months)	(6 Years)
1	*Morus alba*	IC 313703	30.0 (±5.68)	30.0 (±4.71)
2	*Morus bombycis*	EC 493785	66.7 (±2.72)	60.0 (±4.71)
3	*Morus indica*	IC 313703	95.0 (±2.36)	90.0 (±2.36)
4	*Morus indica*	IC 313703	40.0 (±4.71)	40.0 (±4.71)
5	*Morus indica*	IC 313703	90.0 (±4.71)	80.0 (±4.08)
6	*Morus indica*	IC 313703	55.0 (±2.36)	53.3 (±2.72)
7	*Morus indica* × *alba*	IC 313703	60.0 (±2.72)	60.0 (±2.72)
8	*Morus rubra*	EC 493988	25.0 (±2.36)	20.0 (±4.71)
	Average		57.7 (±3.45)	54.2 (±3.84)

## Data Availability

Not applicable.

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
