# Peer review of "Optimized Recovery of Cryostored Dormant Buds of Mulberry Germplasm"

_plants, 2023, doi:10.3390/plants12020225_

Round 1

Reviewer 1 Report

The manuscript “Optimized recovery of cryostored dormant buds of mulberry germplasm”  shows a two-step freezing cryoprotocol preceded by desiccation to 15 to 25% moisture content  and  developed and successfully applied to winter dormant buds of mulberry (different Morus spp.) of core set comprising 238 accessions in laboratory conditions.

The manuscript is prepared professionally. It includes a well-crafted abstract and an exhaustive introduction that justifies the research undertaken. The introduction points to the deficiencies in the literature on the subject. The aim is clearly defined. Modern analytical methods were used in the research. The discussion of the results is well prepared. The conclusions are well-defined. The illustrative material is appropriate.

In my opinion, the manuscript after corrections, will be suitable for publication in a journal.

Detailed comments:

Abstract: Should be included some numeric data obtained from results.

Line 14. please correct as 238 accessions studies.

In abstract several times authors used developed, tested, analyzed and should be one sentence showing what did you do in the study. It seems several aims and it is not good for effective abstract. Please clearly indicate your aims in a single sentence

Line 21. Please delete untreated

Line 26. Please correct as Maximum shoot formation (100%) was obtained from Morus alba. 

Introduction - The introduction is enough in my opinion. Introduction needs an introduction sentence showing importance less-known horticultural species including mulberries

I prepared below one with some fresh references

More recently there were an increasing interest to less-known horticultural plant species including mulberries and the main reason of the interest is that they are among the healthiest foods in horticultural crops. They’re delicious, nutritious, and provide a number of impressive health benefits. Less-known fruits are accepted as insurance in the future horticultural cultivation in the climate change scenario due to their resistance to abiotic and biotic conditions (Dogan et al., 2014; Ikinci et al., 2014; Nadeem et al., 2018; Grygorieva et al. 2021; Juric et al. 2021).

Nadeem, M.A.; Habyarimana, E.; Çiftçi, V.; Nawaz, M.A.; Karaköy, T.; Comertpay, G. et al. Characterization of genetic diversity in Turkish common bean gene pool using phenotypic and whole-genome DArTseq-generated silicoDArT marker information. PLoS ONE, 2018, 13(10), e0205363. 

Grygorieva, O.; Klymenko, S.; Kuklina, A.; Vinogradova, Y.; Vergun, O.; Sedlackova, V.H.; Brindza, J.Evaluation of Lonicera caerulea L. genotypes based on morphological characteristics of fruits germplasm collection. Turk. J. Agric. For. 2021, 45, 850-860.

Juric, S.; Vlahovicek-Kahlina, K.; Duralija, B.; Maslov Bandic, L.; Nekic, P.; Vincekovic, M. Stimulation of plant secondary metabolites synthesis in soilless cultivated strawberries (Fragaria × ananassa Duchesne) using zinc-alginate microparticles. Turk. J. Agric. For. 2021, 45, 324–334.

Line 34-36 needs some references

Mulberry (Morus L., Family Moraceae) that originated at the foothills of the Himalayas is an economically important woody tree species which is extensively used for rearing silkworms. It is an out–breeding, heterozygous and perennial tree species also used in agro-forestry and horticulture 

Ercisli, S.; Orhan, E. Some physico-chemical characteristics of black mulberry (Morus nigra L.) genotypes from Northeast Anatolia region of Turkey. Sci. Hortic. 2008, 116, 41–46.

Gecer, M.K.; Akin, M.; Gundogdu, M.; Eyduran, S.P.; Ercisli, S.; Eyduran, E. Organic acids, sugars, phenolic compounds, and some horticultural characteristics of black and white mulberry accessions from Eastern Anatolia. Can. J. Plant Sci. 2016, 96, 27–33.

Kostic, D.A.; Dimitrijevic, D.S.; Mitic, S.S.; Mitic, M.N.; Stojanovic, G.S.; Zivanovic, A. Phenolic content and antioxidant activities of fruit extracts of Morus nigra L. (Moraceae) from Southeast Serbia. Trop. J. Pharm. Res. 2013, 12, 105–110.

Line 39. Please complete [2.

The introduction is too short must be increased and I suggest to increase it with writing more about the importance mulberries and cryopreservation.

Table 2 is not looking well and should be rearrange it with better shape.

Figure 4. Please add legend as all photos taken by authors and all photos are original.

Table 5 should be move before discussion 

Author Response

Comment: English language and style are fine/minor spell check required

Reply: English language and spelling have been improved.

Comment: Abstract: Should be included some numeric data obtained from results.

Reply: Numerical data has been added inside the abstract.

Comment: Line 14. Please correct as 238 accessions studies.

Reply: Sentence has been corrected.

Comment: In abstract several times authors used developed, tested, analyzed and should be one sentence showing what did you do in the study. It seems several aims and it is not good for effective abstract. Please clearly indicate your aims in a single sentence

Reply: Thank you for your nice suggestion. We have modified the abstract as per your suggestion.

Comment: Line 21. Please delete untreated.

Reply: Thank you for your suggestion. Abstract has been modified according to the suggestion.

Comment: Line 26. Please correct as Maximum shoot formation (100%) was obtained from Morus alba

Reply: Sentence has been corrected.

Comment: Introduction - The introduction is enough in my opinion. Introduction needs an introduction sentence showing importance less-known horticultural species including mulberries.

Reply: Introduction part has been improved with new references as suggested.

Comment: Line 34-36 needs some references

Reply: New references have been added as suggested.

Comment: Line 39. Please complete [2].

Reply: We have been rectified.

Comment: The introduction is too short must be increased and I suggest to increase it with writing more about the importance mulberries and cryopreservation.

Reply: Introduction part has been modified.

Comment: Table 2 is not looking well and should be rearrange it with better shape.

Reply: Table 2 have been rearranged. It was page setting issue.

Comment: Figure 4. Please add legend as all photos taken by authors and all photos are original.

Reply: Figure Legend has been modified according to the suggestion.

Comment: Table 5 should be move before discussion 

Reply: Table 5 has moved before discussion.

The reviewer and editors comments are reasonable, and we have corrected the MS in accordance with the comments and suggestions. A thorough internal reviews was performed in the whole MS, changes highlighted in Track Change Format supplied MS. We are thankful to learned reviewer for giving critical insights, leading to substantial improvement in the manuscript, we hope the response meets the reviewer and editor approval.

Reviewer 2 Report

The main problem is the use of the terms recovery x viability. Both terms must be precisely defined in the Materials and Methods. Data in the same tables are presented in recovery, eg Table 2, in the other in viability eg 1,3,4,5. These terms are used interchangeably in the title text of figures and tables as opposed to within figures, eg Figure 2 and tables eg Table 1,4 used. In this context, I have a proposal to change the title. The paper is more about viability and less about recovery. Viability prevails in MS.

The vague use of these terms is difficult for the reader to understand.

Suggested corrections

Table 5 should be moved to the Results chapter.

In general, it is not necessary to report viability recovery, and a moisture content in two decimal places, round to one decimal place is sufficient.

The name of plant species should be in Latin eg rows 75,84-89,91,94,160-164,172

Same for in vitro eg  rows 15,22,144,171,172,173,195,253,307, 309, 318

Picture 1 shows MC 16-55, it should be MC 46-55

Row 39 is [2, should be [2]

R 46 exists [4, 5,6,7,8], it should be [4,5,6,7,8].

R 94 there is Recovery, it should be Viability

R 96 there is average, it should be the average, ± SD

R 99 there is Recovery, it should be Viability

R 100 please explain the abbreviations IMC, DMC, DC and AC below the table 

R 125 add with and without rehydration

R 163 there is recovery, it should be viability

R 165 there is the mean, it should be the mean, ± SD

R 197 there is Tyler and Stushnoff [18], it should be Tyler and Stushnoff [15]; where is [18] cited?

R 358 there is conflict of interest.”, it should be conflict of interest.

R 391 should be better formatting 

Author Response

Comment: English language and style are fine/minor spell check required
Reply: English language and spelling have been improved.

Comment: Table 5 should be moved to the Results chapter.

Reply: Table 5 has moved in Results part.

Comment: In general, it is not necessary to report viability recovery, and a moisture content in two decimal places, round to one decimal place is sufficient.

Reply: Thank you for your suggestion. We have done it accordingly.

Comment: The name of plant species should be in Latin eg rows 75,84-89,91,94,160-164,172

Reply: The corrections have been made in side the text as per your suggestions.

Comment: Same for in vitro eg  rows 15,22,144,171,172,173,195,253,307, 309, 318

Reply: The corrections have been made in side the text as per your suggestions.

Comment: Picture 1 shows MC 16-55, it should be MC 46-55

Reply: Thank you for your nice suggestion. The corrections have been made in side the figure as per your suggestions.

Comment: Row 39 is [2, should be [2]

Reply: The corrections have been made in side the text as per your suggestions.

Comment: R 46 exists [4, 5,6,7,8], it should be [4,5,6,7,8].

Reply: The corrections have been made in side the text as per your suggestions.

Comment: R 94 there is Recovery, it should be Viability

Reply: The corrections have been made in side the text as per your suggestions.

Comment: R 96 there is average, it should be the average, ± SD

Reply: The corrections have been made in side the text as per your suggestions.

Comment: R 99 there is Recovery, it should be Viability

Reply: The corrections have been made in side the text as per your suggestions.

Comment: R 100 please explain the abbreviations IMC, DMC, DC and AC below the table

Reply: The full form of the abbreviations have been given below the table as per your suggestions.

Comment: R 125 add with and without rehydration

Reply: The corrections have been made in side the text as per your suggestions.

Comment: R 163 there is recovery, it should be viability

Reply: The corrections have been made in side the text as per your suggestions.

Comment: R 165 there is the mean, it should be the mean, ± SD

Reply: The corrections have been made in side the text as per your suggestions.

Comment: R 197 there is Tyler and Stushnoff [18], it should be Tyler and Stushnoff [15]; where is [18] cited?

Reply: Thank you for your nice correction. It was mistake we have corrected accordingly and reference number 18 has been added inside the text.

Comment: R 358 there is conflict of interest.”, it should be conflict of interest.

Reply: The corrections have been made in side the text as per your suggestions.

Comment: R 391 should be better formatting.

Reply: The corrections have been made in side the text as per your suggestions.

The reviewer and editors comments are reasonable, and we have corrected the MS in accordance with the comments and suggestions. A thorough internal reviews was performed in the whole MS, changes highlighted in Track Change Format supplied MS. We are thankful to learned reviewer for giving critical insights, leading to substantial improvement in the manuscript, we hope the response meets the reviewer and editor approval.

Reviewer 3 Report

On my opinion, one of the main critical point of this study is that the innovative aspects of the contribution in respect to some previous published studies are not adequately and accurately put in evidence. These previous studies (below listed) on application of cryopreservation of dormant buds of mulberry germplasm are included and cited but should have accurately considered and described either in the introduction and in the discussion to put in evidence which are the main point of optimization of the protocols in respect to them

1)      Rao, A.A.; Chaudhury, R.; Kumar, S.; Velu, D.; Saraswat, R.P.; Kamble, C.K. Cryopreservation of mulberry germplasm core 374 collection and assessment of genetic stability through ISSR markers. International J. Industrial Entomol. 2007; 15(1), 23-33. 375 9.

2)      Rao, A.A.; Chaudhury, C.; Malik, S.K.; Kumar, S.; Ramachandran, R.; Qadri, S.M.H. Mulberry biodiversity conservation 376 through cryopreservation. In Vitro Cell. Develop. Biol. -Plant, 2009; 45(6), 639-649

Other points:

Lines 83 and 84: Maximum viability (100%) was found after cryopreservation in the accessions of Morus indica x M. alba (IC 313836).

This value, according to the statistical analysis reported in Tab.1 is not significantly different from the value of  Morus indica IC 313703.

Same problem for “Minimum viability (25%) was recorded in M. rubra (EC 493988) and followed by M. latifolia (EC 493831).

Same problem for line 161: while minimum recovery was 12% in M. alba. This value (see table 4) is not statistically different from that of Morus bombycis

Figure 2 does not give, on my opinion, further and clearer information in respect to tab. 1. I suggest adding to tab. 1 a line with ANOVA results considering as source of variation the species.

Data in table 3: would be very interesting to find out though the results of statistical analysis which are the best treatment to be applied to each species to obtain the highest recovery rates for dormant buds. Thus, I suggest performing and show results of this further statistical analysis.

I strongly suggest moving table 5 in results.

Other minor points:

                Figure 1: January instead of Januray

the names of the species should be always in italics (see for instance lines 80-89

On my opinion, one of the main critical point of this study is that the innovative aspects of the contribution in respect to some previous published studies are not adequately and accurately put in evidence. These previous studies (below listed) on application of cryopreservation of dormant buds of mulberry germplasm are included and cited but should have accurately considered and described either in the introduction and in the discussion to put in evidence which are the main point of optimization of the protocols in respect to them

1)      Rao, A.A.; Chaudhury, R.; Kumar, S.; Velu, D.; Saraswat, R.P.; Kamble, C.K. Cryopreservation of mulberry germplasm core 374 collection and assessment of genetic stability through ISSR markers. International J. Industrial Entomol. 2007; 15(1), 23-33. 375 9.

2)      Rao, A.A.; Chaudhury, C.; Malik, S.K.; Kumar, S.; Ramachandran, R.; Qadri, S.M.H. Mulberry biodiversity conservation 376 through cryopreservation. In Vitro Cell. Develop. Biol. -Plant, 2009; 45(6), 639-649

Other points:

Lines 83 and 84: Maximum viability (100%) was found after cryopreservation in the accessions of Morus indica x M. alba (IC 313836).

This value, according to the statistical analysis reported in Tab.1 is not significantly different from the value of  Morus indica IC 313703.

Same problem for “Minimum viability (25%) was recorded in M. rubra (EC 493988) and followed by M. latifolia (EC 493831).

Same problem for line 161: while minimum recovery was 12% in M. alba. This value (see table 4) is not statistically different from that of Morus bombycis

Figure 2 does not give, on my opinion, further and clearer information in respect to tab. 1. I suggest adding to tab. 1 a line with ANOVA results considering as source of variation the species.

Data in table 3: would be very interesting to find out though the results of statistical analysis which are the best treatment to be applied to each species to obtain the highest recovery rates for dormant buds. Thus, I suggest performing and show results of this further statistical analysis.

I strongly suggest moving table 5 in results.

Other minor points:

                Figure 1: January instead of Januray

the names of the species should be always in italics (see for instance lines 80-89

Author Response

Comment: On my opinion, one of the main critical point of this study is that the innovative aspects of the contribution in respect to some previous published studies are not adequately and accurately put in evidence. These previous studies (below listed) on application of cryopreservation of dormant buds of mulberry germplasm are included and cited but should have accurately considered and described either in the introduction and in the discussion to put in evidence which are the main point of optimization of the protocols in respect to them.

1)      Rao, A.A.; Chaudhury, R.; Kumar, S.; Velu, D.; Saraswat, R.P.; Kamble, C.K. Cryopreservation of mulberry germplasm core 374 collection and assessment of genetic stability through ISSR markers. International J. Industrial Entomol. 2007; 15(1), 23-33. 375 9.

2)      Rao, A.A.; Chaudhury, C.; Malik, S.K.; Kumar, S.; Ramachandran, R.; Qadri, S.M.H. Mulberry biodiversity conservation 376 through cryopreservation. In Vitro Cell. Develop. Biol. -Plant, 2009; 45(6), 639-649

Reply: The corrections have been made in side the text as per your suggestions.

Other points:

Comment: Lines 83 and 84: Maximum viability (100%) was found after cryopreservation in the accessions of Morus indica x M. alba (IC 313836). This value, according to the statistical analysis reported in Tab.1 is not significantly different from the value of  Morus indica IC 313703. Same problem for “Minimum viability (25%) was recorded in M. rubra (EC 493988) and followed by M. latifolia (EC 493831).

Reply: Thank you for your input and query about the viability of these germpalsm. As Morus species under in vivo and in vitro condition indicated species-specific variation and most of the wild Morus species were found very sensitive .The accession IC 313836 (Morus indica x M. alba) and IC 313703 (Morus indica ) are not significantly different to each other or we can say that these accessions showing highest viability due to these nondormant nature. Both the accesioons are indigenous and showing maximum viability. However, accession number EC 493988 (M. rubra)  and EC 493831 (M. latifolia ) are exotic germplasm and showing minimum viability due to its dormant nature so that not having much significantly different to each other.

Comment: Same problem for line 161: while minimum recovery was 12% in M. alba. This value (see table 4) is not statistically different from that of Morus bombycis.

Reply: These two accessions belong to different Morus species were also exotic germplasm and the results showed less viability after modifications.

Comment: Figure 2 does not give, on my opinion, further and clearer information in respect to tab. 1. I suggest adding to tab. 1 a line with ANOVA results considering as source of variation the species.

Reply: As in Table1, we used different 19 accessions belongs to 13 diverse Morus species for knowing the state of the dormant buds as it is dormant or semi dormant or Non dormant, while in Figure 2 we showed the Viability of dormant buds of different 13 Morus species before and after cryostorage.

Comment: Data in table 3: would be very interesting to find out though the results of statistical analysis which are the best treatment to be applied to each species to obtain the highest recovery rates for dormant buds. Thus, I suggest performing and show results of this further statistical analysis.

Reply: In the table 3, we described the different combinations used for the optimization of recovery growth of dormant buds after cryopreservation. Now table 3 has been modified according to your suggestion.

Comment: I strongly suggest moving table 5 in results.

Reply: Thank you for your correction. The corrections have been made in side the text as per your suggestions. Table 5 have been moved to result section.

Other minor points:

Comment: Figure 1: January instead of Januray

Reply: The corrections have been made in side the text as per your suggestions.

Comment: The names of the species should be always in italics (see for instance lines 80-89

Reply: The corrections have been made in side the text as per your suggestions.

The reviewer and editors comments are reasonable, and we have corrected the MS in accordance with the comments and suggestions. A thorough internal reviews was performed in the whole MS, changes highlighted in Track Change Format supplied MS. We are thankful to learned reviewer for giving critical insights, leading to substantial improvement in the manuscript, we hope the response meets the reviewer and editor approval.

Round 2

Reviewer 3 Report

The revised version, on my opinion, is suitable for publication.

I suggest only very few minor points on the uploaded file for improving the English version. 

Author Response

Comment: Reviewer suggested only very few minor points on the uploaded file for improving the English version at line number 39, 41, 161, 193, 195, 305 and 306.

Reply: Thank you for suggestion. We have corrected and modified according to your kind suggestions inside the text.

The reviewer and editors comments are reasonable, and we have corrected the MS in accordance with the comments and suggestions. A thorough internal reviews was performed in the whole MS, changes highlighted in Track Change Format supplied MS. We are thankful to learned reviewer for giving critical insights, leading to substantial improvement in the manuscript, we hope the response meets the reviewer and editor approval.
